# Global thermal spring distribution and relationship to endogenous and exogenous factors

G. Tamburello [1] ✉, G. Chiodini [1], G. Ciotoli [2,3], M. Procesi [3], D. Rouwet[1], L. Sandri [1], N. Carbonara[4] & C. Masciantonio [4]

Here we present digitization and analysis of the thermal springs of the world dataset compiled by Gerald Ashley Waring in 1965 into a collection of analog maps. We obtain the geographic coordinates of ~6,000 geothermal spring areas, including complementary data (e.g., temperature, total dissolved solids, flow rate), making them available in electronic format. Using temperature and flow rate, we derive the heat discharged from 1483 thermal spring areas (between ~$10^{-5}$ and ~$10^3$ MW, with a median value of ~0.5 MW and ~8300 MW in total). We integrate this data set with other global data sets to study the relationship between thermalism and endogenous and exogenous factors with a supervised machine learning algorithm. This analysis confirms a dominant role of the terrestrial heat flow, topography, volcanism and extensional tectonics. This data set offers new insights and will boost future studies in geothermal energy exploration.

Geothermal waters represent one of the expressions of the upward energy flow toward surface of the heat stored in Earth's interior. In particular, thermal springs are water discharges whose temperature is sensibly higher than the average external temperature in the surrounding area. In this framework, Pentecost et al.[1] proposed a more constraining analysis of the definition of the term "thermal springs" and suggested that it is reasonable to consider "thermal" spring waters characterized by a temperature higher than the local mean annual air temperature. This implies that spring waters can be thermal even with a temperature slightly above zero Celsius degrees at high latitudes or high altitudes. Obviously, this may create particular cases where a thermal spring could have a temperature far lower than the air temperature during summer[1].

Thermal springs for bathing and health represent one of the most ancient uses of the geothermal resource. Balneology has a long history: the Greeks, Turks, Romans and Japanese were famous for their Thermal-spas[2]. However, thermal springs are also deeply used for other direct geothermal uses such as building heating, greenhouse heating, aquaculture, fruit and vegetable drying and other industrial processes[3]. In the early 2000s, Lund and Freeston[4] estimated that the extracted thermal power was ~15,145 MW, involving >52,746 kg s$^{-1}$ of fluids, and using 190,699 TJ yr$^{-1}$ of thermal energy. However, Limberger et al.[5] have shown that there is a much larger global geothermal resource base in sedimentary aquifers for direct heat use. The global geothermal resource base would range between 125 and 1793 EJ yr$^{-1}$, with a total effective aquifer volume ranging from $4.0 \cdot 10^6$ km$^3$ to $22.8 \cdot 10^6$ km$^3$. The mean heat flow through the total aquifer-overlain surface is 64 mW m$^{-2}$ with a mean aquifer geothermal gradient of 32 °C km$^{-1}$. In the last decades, this enormous potential for direct geothermal heat from aquifers attracted special attention, in particular toward those thermal springs indicating areas in which exploitation of geothermal energy might be economically feasible also for indirect uses such as electrical power production[6–9]. In this framework, the availability of geochemical data besides the location of thermal spring areas assumes particular importance, especially in the first stages of a geothermal exploration program where the level of uncertainty is high, and a reduction of the risk (e.g., wrong estimation of the resource capacity[10]) is required.

[1]Istituto Nazionale di Geofisica e Vulcanologia, sezione di Bologna, Bologna, Italy. [2]Consiglio Nazionale delle Ricerche, Istituto di Geologia Ambientale e Geoingegneria, Rome, Italy. [3]Istituto Nazionale di Geofisica e Vulcanologia, Rome, Italy. [4]Università degli Studi di Bologna, Bologna, Italy. ✉e-mail: giancarlo.tamburello@ingv.it

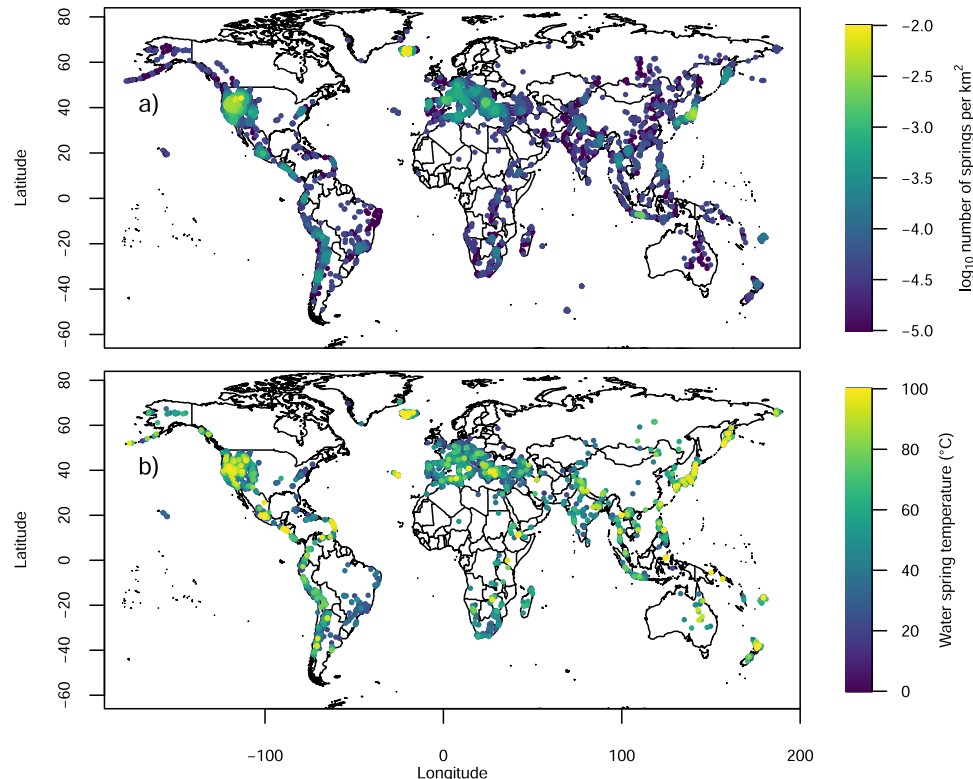

**Fig. 1 | The distribution of thermal springs in the world.** World map of the digitized locations of the global thermal spring areas and their (**a**) calculated kernel density (axis-aligned bivariate normal kernel, evaluated on a square grid of 500 points and bandwidth of 10) and **b** maximum water temperature in Celsius degree if provided by Waring[13] (map created with maps[50] R package).

There are several works (South Italy[10], North America[11], China[12]) where the authors have collected a large data set on thermal springs. One of the most intriguing works of this kind was published by Waring[13] in 1965, who reported an incredibly detailed data set on the thermal springs worldwide. His work describes the distribution and features of more than 6000 geothermal spring areas in more than 100 countries worldwide. The name/location, temperature, flow rate, associated rocks, qualitative/quantitative chemical properties, and references are reported for many thermal springs as tabulated data. Each point illustrated by Waring[13] is meant to be a geothermal area/site that may be characterized by numerous thermal springs. Sometimes this number is reported in the comments, or it is generically written as "several springs". When it is not reported, it is assumed to be a single thermal spring. Unfortunately, the work of Waring[13] misses the numerical notation of the geographical coordinates of each thermal spring area. However, the locations are graphically provided through maps, where each site is identified by a numeric ID that is recalled in the information tables.

In this work, we present a digitized format of the thermal springs of the world of Waring[13]. Our data set contains geographical coordinates (from georeferentiation), temperatures, flow rates and other data. We complement this information with different recent geological data sets available in the literature and analyze them using statistical and geospatial tools and a supervised machine learning algorithm. We show that terrestrial heat flow, topography, volcanism, and extensional tectonic play a key role in the occurrence of thermal waters around the globe.

## Results

### Distribution and temperature of the thermal springs on the Earth's surface

The digitized thermal spring sites from Waring[13] are shown in Fig. 1 (Supplementary data 1). The broad spatial coverage reaches remote locations of the globe. The maps at high latitudes show a high distortion level and a likely more significant error in the georeferencing process. However, most thermal springs are located at lower latitudes, and in this case, the overall error is negligible. We performed an Average Nearest Neighbor Analysis to measure the distances between each digitized thermal spring location (~1500) and the closest (and more precise) location of recently mapped thermal springs in Italy[10] and Northern America[11]. The obtained results revealed a mean distance of ~14 km (~0.125 degrees, Supplementary Fig. 3). We are aware that this calculation assumes that the nearest digitized and recently mapped thermal springs are the same and that this assumption may be erroneous for some springs (e.g., Waring[13] reports thermal springs in Colorado, New Mexico and Arizona that are not reported by Ferguson and Grasby[11]; in this case, the distances are misleadingly long and refer to springs from other states). However, we are confident that on this large number of springs the median distance represents a good indicator of the precision of our spatial digitization. At a smaller scale, we also find a good match between these data sets in terms of their kernel density and distribution of clusters of thermal springs (Supplementary Fig. 4). Most thermal spring areas are reported in the USA and Europe (Fig. 1a). This heterogeneous distribution may represent a bias due to the abundance of literature on thermal springs in these continents, probably related to the past and present higher interest in geothermal energy, and a higher focus by the Waring[13] study on these regions. Overall, the thermal springs cluster along the active tectonic areas and are more scattered in cratonic areas (Fig. 1). There are some evident data gaps in some regions (e.g., Cameroon Volcanic Line) or a conspicuous underestimation of the number of thermal springs in other regions (e.g., Alps[14]). For this work, we decided to not integrate the data set with complementary and more recent data, as we believe that this onerous goal would need more efforts and time, with a high

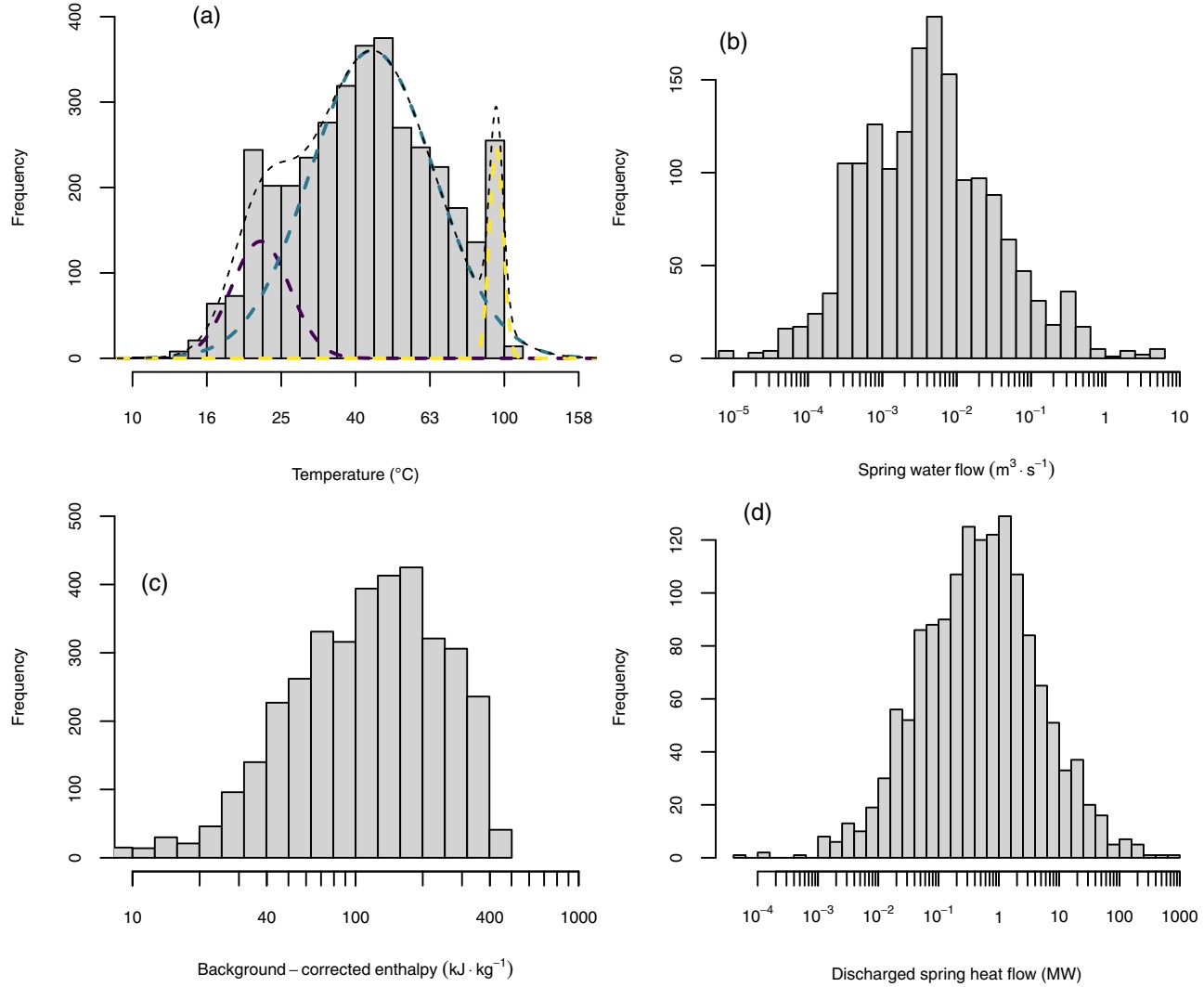

**Fig. 2 | Thermal springs temperatures and enthalpies.** Frequency histograms of (**a**) water spring temperature, (**b**) spring water flow, (**c**) calculated saturated liquid enthalpy and **d** discharged heat. The mix of three normal distributions, dashed lines in (**a**), that likely compose the logarithm of the thermal spring temperature (with antilogarithm means of ~22 °C in dark purple, ~44 °C in turquoise, ~95 °C in yellow, and mixing proportions of ~14%, ~80%, and ~6%, respectively).

probability to remain incomplete somehow. We are confident that our work may represent a useful tool and starting point for a more complete data set.

Water temperatures show higher values along tectonically active margins (Fig. 1b) and lower values in cratonic areas. Temperatures are characterized by a polymodal right-skewed distribution with a median value of 42 °C and an evident buffering at the boiling point (Fig. 2a). Spring flow rates vary from $7 \cdot 10^{-6}$ $m^3 s^{-1}$ to 5.27 $m^3 s^{-1}$ (Fig. 2b).

The digitized data set of Waring[13] provides an overview of the main characteristics of water thermalism worldwide. The logarithm of water temperatures displays three main modes corresponding to ~22 °C, ~44 °C, and ~95 °C (obtained with a Gaussian kernel density estimator with a bandwidth of 3.5 selected with Silverman's rule of thumb) that are also derived as averages with an expectation-maximization (EM) algorithm for normal mixtures[15]. We suspect that lower and upper temperature modes may be biased. Thermal waters with temperatures below 20 °C are unlikely to be reported in most of the climate regions (except in the coldest). Instead, the mode at ~95 °C may derive from the buffer temperature of the boiling process, generating a natural clustering near 100 °C. The mode at ~44 °C is probably less biased and indicates the most frequent temperature of the reported thermal springs. However, it is not surprising that the boiling/near-boiling thermal waters are entirely located along active margins.

Although chemical analysis is only occasionally reported, the available data yet allows us to explore some relevant dynamics. The Na-Cl "mixing" plot (Fig. 3a) shows how the chemical composition of the thermal waters ranges from a water-rock interaction domain to a more seawater-controlled domain, in particular for the thermal springs near coastlines (given by the smaller size of the circles in Fig. 3). The Ca-SO$_4$ plot (Fig. 3b) shows the possible interactions with anhydrite (aligned along the stoichiometric anhydrite/gypsum line), the effect of calcium precipitation (low $Ca^{2+}/SO_4^{2-}$) or carbonate rock interaction (high $Ca^{2+}/SO_4^{2-}$), and interaction with and formation of steam-heated waters. The Gibbs diagram[16] is a standard tool to establish which key process controls the water chemistry. The plot is divided into three distinct areas dominated by precipitation, evaporation and rock–water interaction. Most of the waters fall between the rock–water interaction dominance and evaporation dominance fields. The $Cl^-/(Cl^-+HCO_3^-)$ ratio allows discriminating a group of waters in which there may be an essential contribution of endogenous $CO_2$ (higher $HCO_3^-$ concentrations, hence lower $Cl^-/(Cl^-+HCO_3^-)$ ratios).

The $Cl^--SO_4^{2-}-HCO_3^-$ ternary diagram[17] (Fig. 3d) is commonly used to classify thermal waters based on three major anions. This is because these three anions are robust markers of different sources and mechanisms: $Cl^-$ comes from seawater and/or processes of mixing

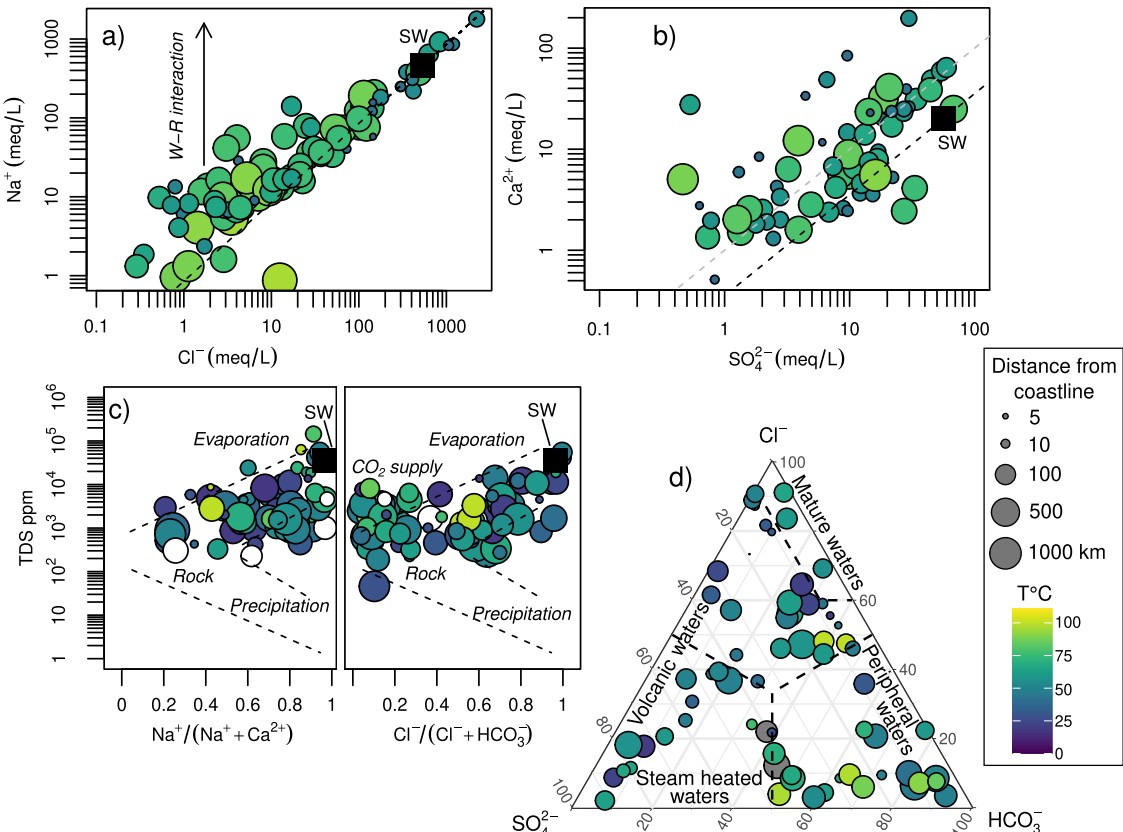

**Fig. 3 | The chemical compositions of thermal waters.** Chemical compositions of thermal waters in relation to their temperature (pseudo-color scale) and their distance from the nearest coastline (size of the circles). **a** Relationship between $Cl^-$ and $Na^+$ (in meq/L), showing that samples range from a water-rock interaction (W-R) domain to seawater (SW) domain. **b** The $SO_4^{2-}$ and $Ca^{2+}$ relationship shows the possible interaction of thermal waters with anhydrite and seawater. **c** The Gibbs diagram shows which are the foremost vital processes (precipitation, water-rock interaction, evaporation) controlling the chemistry of these thermal waters. **d** The ternary $Cl^-$-$SO_4^{2-}$-$HCO_3^-$ ternary plot (Giggenbach, 1991) shows the different types of thermal waters.

(with deep geothermal fluids) and evaporation; $SO_4^{2-}$ comes from the oxidation of sulfide in volcanic-hydrothermal fluids (only a small amount comes from seawater) and from interaction with sulfate rocks; $HCO_3^-$ comes from volcano-tectonic degassing structures or biological sources. Despite the scarcity of water compositions in our data set, the available data covers all the types of waters represented by the ternary diagram: mature waters, in which long water residence time leads to water-rock equilibrium conditions and Na-Cl enrichment; peripheral waters, dominated by deep-sourced $CO_2$ dissolution at low temperatures; steam-heated waters, characterized by high temperatures and acidity from oxidation of $H_2S$ to $H_2SO_4$; volcanic waters, which cover the evolutionary trend between the premature steam-heated waters and the aged mature waters.

### Heat discharge from thermal springs

For each thermal spring site, we obtained the background-corrected saturated liquid enthalpy (in $kJ\,kg^{-1}$). We calculated the difference between the saturated liquid enthalpy of the spring at its temperature ($H_{spr}$) and the saturated liquid enthalpy of the water at the average ambient temperature[18] in the area of the spring ($H_{amb}$), which is from now on referred to as "background-corrected enthalpy" ($H_{spr}$ - $H_{amb}$). The saturated liquid enthalpy has been derived using the IAPWS95[19] package for R developed to calculate thermophysical properties of water and steam. Hence, we calculated the heat discharged for the springs reporting both temperature and flow rate with the following equation:

$$Q_H = (H_{spr} - H_{amb}) \cdot V_f \qquad (1)$$

$Q_H$ is the spring heat flow in MW, $H_{spr}$ - $H_{amb}$ has defined above as "background-corrected enthalpy", and $V_f$ is the spring flow rate (converted from $m^3\,s^{-1}$ to $kg\,s^{-1}$, assuming a standard water density). In Fig. 4, we show the global distribution of the thermal springs and their calculated enthalpy. The ambient temperature (background) correction allowed for the discrimination between the thermalism in cratonic (e.g., eastern north and south Americas) and tectonically active (e.g., the Circum-Pacific Belt) areas, respectively characterized by low and high spring enthalpies.

The heat flow of the thermal springs worldwide shows a median value of ~0.5 MW and reaches a maximum value of ~1000 MW (Fig. 2d). This median value is very close to the average net flux calculated by ref. [14] for the thermal springs of the Alps (0-4–0.7 MW, for 210 springs with available temperature and discharge rate). The total heat flow produced by the thermal springs of this catalogue is ~8300 MW. This total value is probably too low and far from a global thermal spring heat flow due to the incompleteness of the data set (e.g., the total absence of thermal springs in the Cameroon Volcanic Line). For the thermal springs of the Alps, Luijendijk et al.[14] reported a total heat flow of 84–146 MW. For the same region Waring[13] reported ~100 MW. As already suggested by some previous research[20–22] for large areas of the Cascade Mountains in the United States and for central Apennine springs in Italy, our results confirm that springs characterized by high volumetric discharge rates and low temperatures may transport a substantial amount of the crustal heat flow. This is evident in the spring flow rate vs background-corrected enthalpy in Fig. 5, where we show isolines of discharged heat (in MW). This plot also shows a noticeable clustering of the majority of thermal springs around ~1 MW (see also histogram plot in Fig. 2d) and determines two

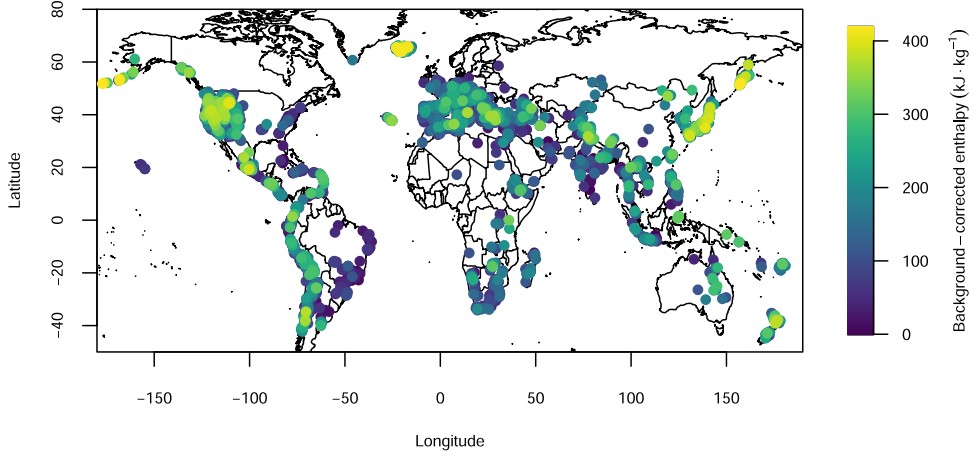

**Fig. 4 | Thermal springs background-corrected enthalpies.** Map showing world distribution of calculated background-corrected saturated liquid enthalpy (in kJ kg⁻¹) for 3680 thermal spring areas (map created with maps[50] R package).

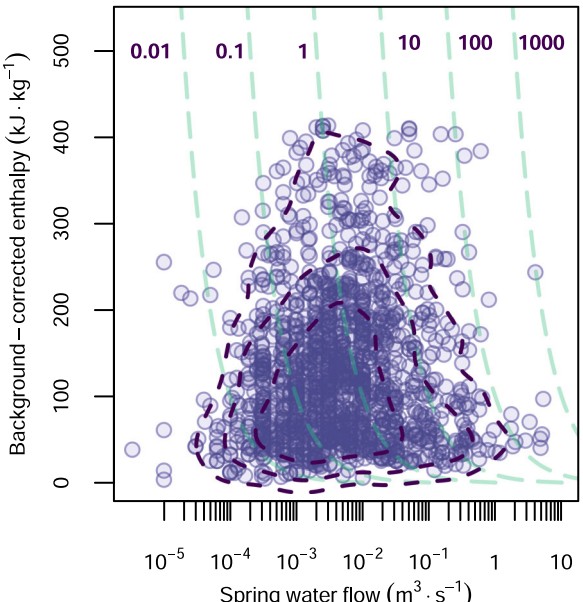

**Fig. 5 | Background-corrected enthalpy vs spring water discharge rate.** Relationship between background-corrected saturated liquid enthalpy and spring water discharge obtained in this work. The turquoise dashed isolines represent different values of discharged heat in MW. The dark purple dashed lines enclose 95%, 80% and 50% of the reported thermal springs.

main intervals, at low and high water flows, respectively, in which thermal springs tend to have lower background-corrected saturated liquid enthalpies. Ferguson and Grasby[11] explained this trend considering the role that flow system geometry (expressed in terms of the depth-to-length ratio of the flow system) plays in advection in groundwater flow systems. In typical geological settings[23], advection is observable at an optimal depth-to-length ratio of ~0.1. Higher and lower values may be unfavorable to the formation of high-enthalpy thermal waters within the range of terrestrial heat flow observed for continents[24] ($67 \pm 31$ mW m⁻²). This explanation is also reinforced by the following simple consideration: low spring water flows facilitate thermal exchange between groundwater and the surrounding rock. We find a slight but significant correlation (Pearson correlation coefficient 0.15, $p$ value $3.27 \cdot 10^{-8}$ in a test for association between paired samples) between the logarithms of the calculated springs heat flow and their corresponding mean terrestrial heat flow[24]. This modest

correlation provides evidence of the critical role that terrestrial heat flow plays in controlling the occurrence of water thermalism. A more robust confirmation is provided in the next section.

**The role of geological factors on thermal waters occurrence**

Here we use a random forest machine-learning algorithm[25], as described in the Method section, to predict the number of thermal springs per unit area, i.e., the abundance of thermal springs across the globe in relation to several geological factors (Supplementary Data 3). The boxplot chart in Fig. 6 shows the statistical distribution of the calculated variable importance obtained by each of the 500 runs. The variable importance is defined as the relative influence of a single geological factor on the random forest analysis. We carried out two different analysis considering (i) the number of geothermal sites (Fig. 6a), and (ii) the number of thermal springs (Fig. 6b) in each hexagonal cell. Of the 16 geological factors we considered for this analysis, four of them seem to significantly influence the density of thermalism worldwide: terrestrial heat flow[24], topography[26], volcanism[27] and extensional tectonic[28]. The importance of the first two geological factor differs depending on the predicted variable we use. Terrestrial heat flow is dominant if we consider the number of geothermal spring areas (Fig. 6a); instead, topography is dominant if we use the number of thermal springs in each area (Fig. 6b). Terrestrial heat flow is one of the primary drivers for the geothermal springs as it provides the necessary energy for heating. The terrestrial heat flow in the areas covered by the thermal springs is $82 \pm 19$ mW m⁻². The fact that topography (meant as topographic position index[26], or TPI) plays a relevant role in the occurrence of thermal springs and, in general, of groundwater systems is not surprising as it has been already discussed in many previous works[29,30]. Higher volcanic density increases the chance of thermalism associated with the presence of magmatic bodies. Hence, normal faulting (extensional tectonic) may facilitate the formation of pathways for the rising hot fluids toward the surface, thus enhancing thermal advection. A similar conclusion is reached for the distribution of $CO_2$-rich springs and tectonic degassing worldwide[31]. Crustal thickness[32] displays a large variability of its importance factor, especially with the number of geothermal spring areas (Fig. 6a), pointing to a strong dependence on the selection process of the training data set. The crustal thickness underneath the geothermal springs of this data set is $34 \pm 9$ km, slightly thinner than the $39 \pm 6$ km thickness of the continental crust. This may suggest that thermalism is facilitated in regions with thinner crust. However, in light of the wide dispersion of the importance of crustal thickness toward low values, this conclusion must be considered with caution.

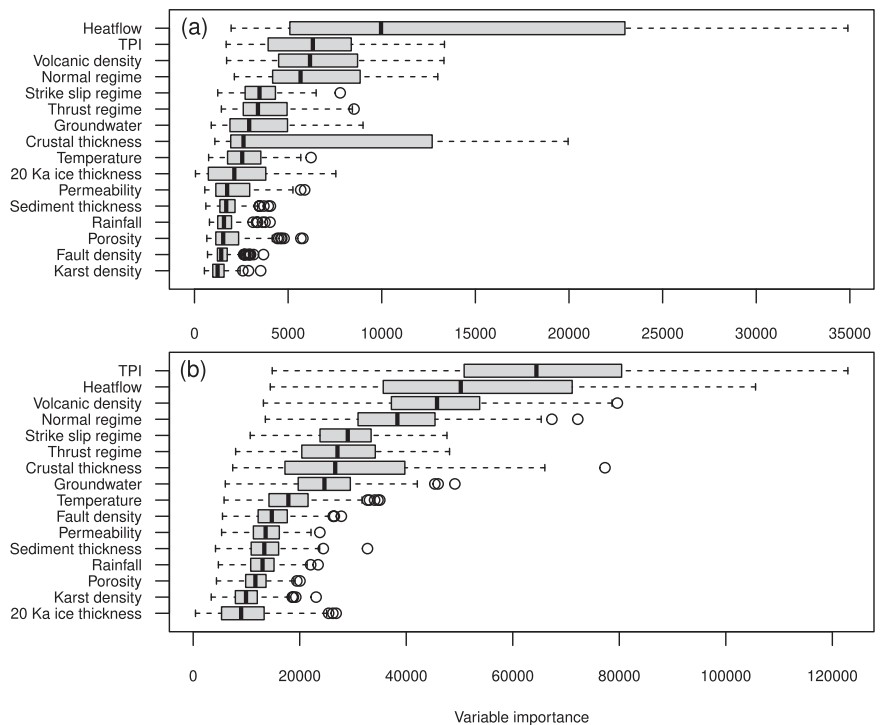

**Fig. 6 | Geological factors importance in thermal springs.** Boxplots of variable importance resulting from the Random Forest algorithm run on 500 randomly sampled data sets to predict the (**a**) number of geothermal spring areas and **b** number of thermal springs.

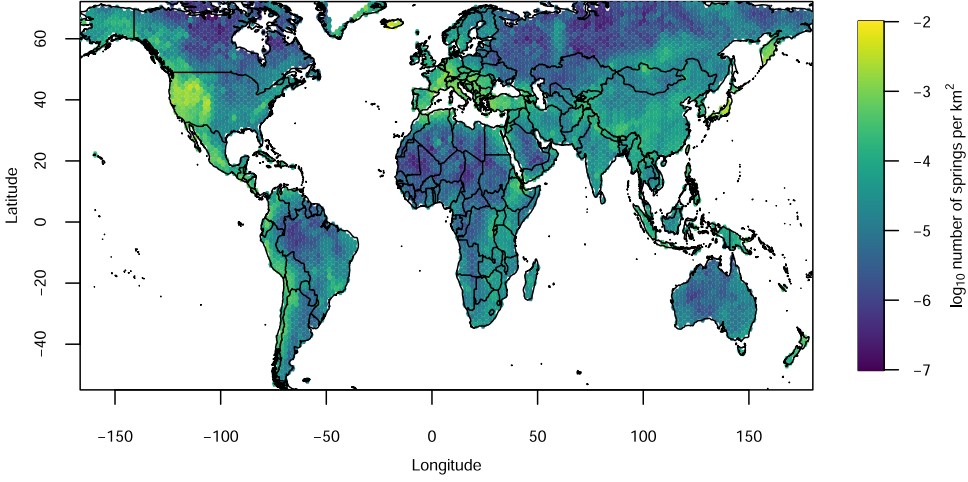

**Fig. 7 | Predicted thermal spring densities in the world.** Random forest prediction of the distribution of the number of thermal springs across the globe. The value represents the averaged 500 random forest results divided by the area (in km²) of each hexagon (map created with maps[50] R package).

The model obtained with the random forest algorithm can be used to predict the number of thermal spring areas (Fig. 6a) and thermal springs (Fig. 6b) in each cell of the hexagonal grid. The predicted number of thermal springs seems to better enhance the contrast between more and less geodinamically active regions (Fig. 7, the predicted number of thermal spring areas is shown in Supplementary Fig. 7). If we compare our results (Fig. 7) with the initial thermal spring distribution (Fig. 1), we can see areas where the number of thermal springs may be much higher (e.g., Cameroon[33], Tanzania[34], Malawi[35] and Mozambique[36], see also Supplementary Fig. 8 and Supplementary data 4). However, further investigations are required on the reasons why the algorithm erroneously predicted a higher density of thermal springs in other regions, such as the Scandinavian and western Australia, where there are very few or no thermal springs.

## Future directions

In this work, we make available to the scientific community the formidable literature review of Gerald Ashley Waring on the geothermal springs of the world. This data set in electronic format will be beneficial for future research on the spatial distribution of thermalism at a small scale and the variation of temperature and flow rate of several thermal springs in the last decades in certain regions. We test its potential by combining it with recent geological data sets. We calculate the discharged heat for 1483 thermal springs (median value ~1 MW) and use a supervised machine learning algorithm to understand which geological factors play a major role in determining the spatial distribution of the thermal springs in the world. We find that, in order of importance, terrestrial heat flow, topography, volcanism, and extensional tectonic are the primary factors. These results can be also extremely useful to

**Table 1 | List of the complementary data set used in this work**

| Name | Type | Reference |
|---|---|---|
| Thermal springs of the world | Raster | [13] |
| Thermal springs in North America | Data Table | [11] |
| Thermal springs in Southern Italy | Data Table | [10] |
| Heat flow | Data table | [24] |
| Holocene volcanoes | Data table | [27] |
| Earth crust thickness | Raster | [32] |
| Seismic stress | Data table | [28] |
| Precipitation and temperature | Raster | [18] |
| Tectonic structures | Vector | [45] |
| Groundwater volume | Raster | [38] |
| Porosity and permeability | Vector | [39] |
| Karst aquifer | Vector | [47] |
| Topographic position index | Raster | [26] |
| Ice thickness history | Raster | [44] |

address the geothermal interest toward specific and less studied areas and especially to drive the first steps of the geothermal surveys and furthermore detailed investigations. Finally, we hope that this work may boost and facilitate future initiatives on the creation of an updated global data set of geothermal springs.

## Methods

### Maps and thermal springs digitization

In this study, we used a software tool (the Georeferencer tool in Quantum GIS[37]) to digitize and extract the geographic information of the geothermal springs reported in Waring[13]. Meridians and parallels intersections and known geographical features (e.g., cities, promontories) reported in the figures allowed us to assign known longitude and latitude values to the selected pixel coordinates ($X_P$, $Y_P$). The three-dimensional $X_P$-$Y_P$-longitude and $X_P$-$Y_P$-latitude points have been fitted with a Thin Plate Spline algorithm, which can introduce local deformations in the data and is useful when very low-quality originals are being georeferenced, such as the old maps in the figures of Waring[13]. This procedure stretches the map assigning a longitude and latitude value to all the pixels in the image (Supplementary Fig. 1). Hence, each thermal spring location is manually derived by selecting it with a mouse cursor. We retrieved a total of 6091 coordinates, among which 750 are with the same position (i.e., some thermal springs referred to the same site).

When available, we included in the data set the total number of thermal springs in each site (ranging from 1 to 150 springs per area, for a total of ~12,500 springs, marked >1 when described as "several springs"), flow rate and temperature values (~27% and ~62% of the mapped spring areas, respectively) and converted them to $m^3 s^{-1}$ and °C. Temperatures are given as ranges or single values. We considered the maximum temperature of the springs in the data analysis. Total dissolved solids content (TDS, expressed in ppm) is provided for 1023 thermal springs. Quantitative chemical analysis of major elements is also available (in ppm vol.) but only for a few hundred thermal springs and not always for the same elements (Supplementary Fig. 2). The most common anions are chlorine, sulfate and bicarbonate, while cations are calcium and sodium. We added information on the presence of resorts (commercially developed springs at spas and health resorts) and if the water is used for bathing. The presence of fumarolic activity is observed in 139 sites.

### Machine learning analysis

For a more exhaustive analysis of this data set, we complemented our retrieved geothermal water dataset with other global data sets available in the literature and described below. We tried to select the most

critical exogenous and endogenous factors that may influence thermalism on Earth. Among the exogenous variables, the precipitation controls the amount of water recharging the local aquifers. The historical climate data set (1970–2000)[18] provided the average temperature and precipitation. The volume and distribution of the modern groundwater (last 50 years)[38] recharged by global precipitation have recently been estimated by analyzing different data sets, among which porosity and permeability[39] that we have also considered in our work. Hubbert[29] and Toth[30] defined the water table in regional groundwater systems as a subdued version of topography. Despite this being not always the dominant control on groundwater levels[40], we added the topography among the exogenous factors. In particular, we considered the global topographic position index[26], a parameter useful to measure topographic slope positions and to automate landform classifications. Regions de-glaciated in the last 50ka are characterized by low present-day temperatures and a low geothermal gradient in the shallow subsurface[41–43]. We argue that this condition may have an effect on the distribution of geothermal springs in these regions. A recent updated model (ICE-7G)[44] provides the ice thickness every 1 ka year since 26 ka. For our study we selected the ice-thickness at 20 ka (the Last Glacial Maximum, LGM).

Among the endogenous factors, heat flow plays a crucial role in representing the upward energy flux that an aquifer can potentially capture, thus increasing its temperature through a unit surface. An updated global heat flow map is provided[24], based on new direct measurements and additional geological and geophysical information for a prediction on a worldwide 0.5° × 0.5° resolution. The proximity of recent magmatic bodies at depth can also increase the temperature of a near aquifer. The position of Holocene and Pleistocene volcanoes is obtained from the Smithsonian Institution data set[27]. Seismic activity is an indicator of current active tectonic processes of the Earth's crust and can be divided into different tectonic regimes or styles of faulting[45]. The World Stress Map project[28] compiled a database of 42,870 earthquakes and their stress regimes distributed across the globe to classify regions with different tectonic styles. Global seismic data has been used to build a model of the Earth's crust[32]. We also considered the sediments and the total crustal thickness among the numerous crustal parameters reported in the Lithos 1.0 model[32]. The presence of tectonic structures (e.g., active and inactive faults) may facilitate the rise of geothermal fluids through the formation of fractures and pathways. Thus, for our analysis we calculated the kernel density of the structural data contained in the geological map of the world[46]. Karst aquifers have unique hydrogeological characteristics, among which high permeability and heterogeneity (complex conduit networks and fractures), or rapid variations of discharge and water table level. Here, we considered the density of karst aquifer worldwide[47] to understand if these particular aquifers play a role in geothermal water occurrence. All data sets used for the purposes of this work are listed in Table 1.

Geospatial and geostatistical analyses of multiple global data sets represent an excellent tool to evaluate the relationship between geological factors and their influence on one particular studied dynamic[48,49]. In this work, we applied the Breiman's random forest algorithm[25]. In particular, for this analysis, we created a hexagonal grid (inradius of 1 degree, 4143 total hexagonal cells on dry land) and calculated the area in $km^2$ for each cell. Hence, we calculated the statistical parameters (minimum, maximum, average and standard deviation) of the data sets of Table 1 for each hexagonal cell (Supplementary Figs. 5–6 and Supplementary Data 2). Finally, we counted for each cell the number of geothermal sites and, in addition, the number of thermal springs (considering that each site may host several thermal springs as mentioned above). Hence, we generated 500 random training data sets of 2000 hexagons and ran 500 random forest analyses for each of these two data tables.

## Data availability

The authors declare that the data supporting the findings of this study are available within the supplementary information files. Coastlines and international boundaries in the figures are created with the R package "maps" (https://cran.r-project.org/package=maps).

## Code availability

The code used for random forest analysis is available at https://www.stat.berkeley.edu/~breiman/RandomForests/ (Fortran) and https://cran.r-project.org/web/packages/randomForest/index.html (R).

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

## Acknowledgements
This work was partially supported by the Italian MIUR grant n. PRIN2017-2017LMNLAW (Connect4Carbon).

## Author contributions
G.T. conceived the idea for this paper and performed the statistical analysis; G.T., N.C., and C.M. digitized the datasets; G.T., G.Ch., G.Ci, M.P., L.S., and D.R. contributed to refinement of the initial concept, and to data interpretation. G.T. drafted the original version of the paper with contributions from all coauthors.

## Competing interests
The authors declare no competing interests.
