## [Peer Review File · Nature Communications]

REVIEWER COMMENTS

Reviewer #1 (Remarks to the Author):

Review of Global Thermal Spring Distribution

This paper on H₂O springs follows the NatureComms paper on global CO₂ degassing of similar authorship and similar style. The success of that CO₂ paper shows how this paper could be similarly successful, but in this case, as yet, there is significantly more work to be done.

My first reaction was, what a fun paper – cleverly constructed, showing the value of classic databases, discussing how to digitize them and how to look for useful correlations. I think the presentation of background-corrected enthalpy (Fig. 4) is particularly sensible; and I certainly applaud the effort to estimate global heat production by advection of thermal waters. But after careful reading I believe there are (1) major additions to the dataset that authors seeking to publish in a Nature journal should expect to make; and (2) overlooked possible correlations to be explored and interpretations to be considered. In addition there is a disappointing level of (3) carelessness in preparing references, data tables, etc. that would have been better addressed before first submission.

My major comments can all be corrected with moderate effort (assimilating new data, re-running the existing codes) and without significant re-structuring of the paper; only after that will this paper be as deserving of publication in NatureComms as the earlier CO₂ paper.

(1) The abstract conveys the impression that the major achievement is digitization of Waring's classic catalog; if that were so this paper would not belong in NatureComms. In fact the importance is (or will be) the analysis of a global data set – but the dataset analyzed here is woefully incomplete. Waring's compilation in 1965 was a tour de force for the time, and it is indeed valuable to utilize this; but so many other digital resources already exist, duplicating and improving on Waring, many created by national geological surveys, that really must be included for a meaningful global analysis. [Line 95 stating "probably, more specific research would discover missing thermal springs" is a patent under-statement: of course it would!]

The authors adopt an unexplained scattershot approach: in addition to Waring they include an unspecified number of data from Cameroon, Svalbard and Deception Island (and fail to provide the reference information for Cameroon and Deception). A cursory web search provides so much more. Waring has less than 50 springs in New Zealand's North Island – but “New Zealand thermal springs chemistry” produced a web-site with >1000 springs as the second hit. If there are 20 times as many springs does this mean that the global production given by the authors for their catalog as 7.8 GW (in the Abstract, or as 8.4 GW at line 221 – why the discrepancy?) should be multiplied by 20? Unfortunately in Figure S6 (which is lacking a caption) the sub-plot of Africa obscures New Zealand so we have no idea whether the authors' random-forest algorithm predicts very many missing springs in New Zealand. The largest area where the authors believe there are more springs to be discovered (Figure S6) is in western China: perhaps unsurprising given this part of the world was largely inaccessible, by politics and by language, to Waring in 1965. But there is now so much more information: my googling “Tibet thermal springs chemistry” produced first a reference to a very recent catalog of over 200 springs, ~10x Waring's compilation; and as the second hit a catalog of 39 more springs in an area where Waring has none. [I had thought that perhaps the authors' reference to Keshi as a “large dataset on thermal springs” might fill the Chinese data gap, but this paper contains no listing of data, which leaves me wondering why was it cited? However, there are other recent compilations in the literature published before the authors submitted their own work.] Although each resource seems small compared with the headline number of >6000 springs in Waring's catalog, that headline number obscures the fact that “only a few hundred springs” have reported chemistry (Figure 3); and <1500 have data enabling calculation of heat flux (Figure 2b, 2d, 5). Surely these statistics can be greatly improved by an additional literature search? Similarly the Minissale and the Ferguson-Grasby datasets (Figure S4) and the East African datasets (Figure S6) should be included in the main analysis. After all this, you can still make predictions of areas of missing data-sets, and now it will be more interesting and more useful!

The authors may argue that a complete literature search would be too onerous, and adding only the most easily found digital datasets would be capricious and statistically unwarranted: but why then do they add Cameroon-Svalbard-Deception to the already geographically biased compilation of Waring? Frankly, the real ‘danger’ is that Waring might prove to be essentially superseded, and all the excellent work the authors did to capture his data prove to be superfluous.

(2) Your variable-importance analysis (Fig. 6) (that allows you to make predictions of spring density in Figure 7) is only as good as your ability to include all the relevant effects, i.e. its validity depends on your imagination, not on the mathematics. To me there are two obvious effects not included in Table 1 (or Fig. S5) that I think would improve your analysis and predictions.

First, local relief! Springs require a pressure differential to manifest at the surface. In principle this could be due to thermally created density differences, but water density changes by <5% between 0 and 100°C, and spring chemistry can increase density by ~4% (using sea-water end-member, Fig. 3). The driving pressure produced by a 1% density change over a 1km hydrostatic head is equivalently

produced by <5m of topography! I predict that high local relief within a few km to a few tens of km will prove to be correlated with larger numbers of springs. (Obviously the most visible thermal manifestations (geysers, fumaroles) are driven by steam pressure, leading to their great visibility and over-representation on Figure 2).

Second, incomplete re-equilibration of geothermal gradients in post-glacial time. Regions de-glaciated in the last 50ka will likely have temperatures decreasing with depth in the shallow sub-surface. I predict that regions still ice-covered at say 20 ka will prove to be correlated with lower numbers of thermal springs. Indeed, in Figure S6 (after western China discussed above), the next largest grey area of predicted under-reported springs is Scandinavia, and then areas of western Canada and Alaska's North Slope. Both are areas of late de-glaciation. And whereas googling New Zealand and Tibet brought up hundreds of springs missed by Waring, my equivalent search for Scandinavia produced ... nothing significant.

I urge you to re-run your random forest algorithm to test these two correlations, local relief and glacial history (of course, on the greatly expanded dataset I suggest you create).

I also suspect you are mis-interpreting the existence of "three main groups: warm, hot and boiling" (Figure 2a). I believe the peak above 90°C, and the sharp fall-off below 20° are biases in reporting. As I noted above, springs which are at local boiling point (or for which local boiling is reached in the very shallow sub-surface) will be visibly steaming and likely effusive or geysering, so will be more widely noted particularly in early reports which are the basis of many of Waring's data. Compilations by national surveys may prove to be more exhaustive and less biased. Springs with temperatures below 20°C are simply unlikely to be reported, except in the coldest climates. From the Pentecost reference the authors cite, "Waring (1965) noted that in Europe, thermal springs would normally be higher than about 20 °C and in the United States, above 15 °C".

My expectation is that, above a lower temperature bound at which the catalog of springs is complete, the number of springs will decrease with increasing temperature, with the mathematical form of that decrease containing information about the relative difficulty of extracting water from different depths. If I am wrong, can you suggest any reason why there should be three separate modes of water temperature?

(3) Miscellaneous errors or comments (all minor, except that their large number raises questions about the authors' rigor)

The .xls file has a lot of spurious false precision (temperatures to the nearest millionth of a degree; heat-flow accurate to 10 picowatts Watts/m²). Please display data appropriately! Lats and longs from Waring to two decimal places, not to the nearest millimeter, with a footnote indicating that median accuracy is likely 0.1°

Table 1 – Heidback and Pasyanos references are reversed

Line 135: 14 km is $\sim 0.125^\circ$, not 0.0002° of latitude!

Waring (1965) is alternately cited as 1976 (Table 1 and Figure 1); and 1995 (line 127 and 147)

References to Tamburello, Le Marechal and Kusakabe are missing (and perhaps others, these were just the ones for which I looked).

Line 67: what is the basis for stating that Waring's maps are "very accurate"?

Georeferentiation should be georeferencing, throughout; neglectable becomes negligible.

Figures 6 and S7 seem to have at least one hexagon in an entirely water-covered area west of Alaska.

Fig 1a – what are units of kernel density? I'm guessing $\#/km^2$? The color scales of Fig. 1 and Fig. 7 need to be made equivalent, not offset by orders of magnitude.

Line 126 et seq. Results, and Line 177 et seq. Discussion are written implying the data plotted in Fig. 1 and Fig. 2 and Fig. 3 are from Waring alone – but in fact I am guessing they also include the Cameroon-Svalbard-Deception data. Please be explicit. Given how few of Waring's data have chemistry, to what extent is Figure 3 dominated by the Cameroon-Svalbard-Deception data? How many data points are these?

Simon Klemperer

Stanford 4/15/22

Reviewer #2 (Remarks to the Author):

This is an excellent contribution and I congratulate the authors for their work. This work will be extremely helpful for geoscientists of many disciplines. The work can be published after some minor revisions outlined below.

Main points:

It is stated in line 173 that gas discharge coordinates are shown in supplemental table data 1 but I did not see any information on gas discharge coordinates in this Table. Here it would be also useful to briefly mention if there are any data on the occurrence of gas discharges from any of these prongs. For example is there evidence for gas bubbles in the springs. If this is not available, in the current data set it could be pointed out that there is this gap in knowledge. Obviously it is beyond the scope of the paper to catalogue gas discharges from springs.

Fig. 3 d is very interesting. Authors state that peripheral waters are usually cool and dominated by deep sourced CO₂. However, in the diagram it appears that these waters have some of the highest temperatures. Likewise the volcanic waters for which we may expect high temperatures show some of the coolest temperatures. More explanation and clarification would be helpful when taking out this figure in the text.

Minor points:

l. 29 'of' not 'about'

l. 51 Better "The aspect of electrical power production"

l. 53 clarify what risk is limited here.

l. 58 delete 'a'

l. 71 'map images'

l. 74 'allowed us to'

l. 81 'among which 750 are with..'

l. 96 better 'to be included in'

l. 131 negligible

I. 149 'values'

I. 149 'and lower values'

I. 227 'majority of '

Point-by-point response to the reviewers' comments

Global thermal spring distribution and relationship to endogenous and exogenous factors

REVIEWER #1

We thank the reviewer for these useful comments. Before replying point by point to each comment, we wish to make a premise. Although Waring dataset is likely far from being a complete dataset, to our knowledge it is currently the only global thermal spring dataset available. We show in our work that it represents a good tool for understanding where most of thermal springs occur in the world and, with appropriate caution (e.g. random re-sampling of ML training dataset), how it can be used for global scale geostatistical analysis. During the initial stage of our work we carried out a short additional literature research (e.g. Cameroon, based on our personal past research experience) and, after the reviewer's suggestion, we tried to extend this research. But the difficulties we encountered have discouraged this work: this operation is really too onerous, it definitely needs much longer time and it requires much more dedicated workforce with an ad-hoc research project. We also think that a further research of complementary data would not be immune from data incompleteness and spatial inhomogeneity among different datasets. Not all the datasets are available as easily accessible data tables, and some of them are fragmented among multiple papers. Thus, we are at a crossroads: we keep looking for new datasets until we cover a good part of the ~ 100 countries reported by Waring, or we keep Waring and we perform a geostatistical analysis by taking into account its low degree of completeness. The latter approach is the one we have finally chosen for this work (consequently, we agreed on removing complementary Cameroon etc. to be consistent), by randomly resampling the Waring dataset for training the random forest algorithm. Moreover, thanks to the second reviewer's suggestion, we added in the dataset how many thermal springs occur at each site. In fact, whereas in the first manuscript version we erroneously referred to each point as a single thermal spring, they are actually geothermal sites that may host several springs (from 1 to 150, sometimes generically described as "several springs"). If we sum up all the thermal springs listed we obtain a total of ~ 12,000. Thus, we re-run our processing considering the number of thermal springs for each site when specified. In our opinion, the results we obtained (the digitized dataset, first-order global distribution of thermal springs in the world, the role of geological factors in geothermality, the statistic on discharged geothermal heat) are important results worth to be shared. We hope that our motivations supporting the results of the paper in its revised version are sufficient to be considered for a

publication on NatComm. We have to say that the quality of our paper has much improved thanks to the reviewers' useful suggestions.

Waring has less than 50 springs in New Zealand's North Island – but “New Zealand thermal springs chemistry” produced a web-site with >1000 springs as the second hit. If there are 20 times as many springs does this mean that the global production given by the authors for their catalog as 7.8 GW (in the Abstract, or as 8.4 GW at line 221 – why the discrepancy?) should be multiplied by 20?

We searched “New Zealand thermal springs chemistry” as suggested by the reviewer. Our second hit is a paper “Microbial biogeography of 925 geothermal springs in New Zealand” that does not contain data tables useful for our work. The other results did not show thermal spring datasets. Searching thermal spring datasets is not so straightforward as suggested by the reviewer and requires much more time, people and efforts beyond our current possibilities. In any case, the GW discrepancy in the text has been corrected.

Unfortunately in Figure S6 (which is lacking a caption) the sub-plot of Africa obscures New Zealand so we have no idea whether the authors' random-forest algorithm predicts very many missing springs in New Zealand.

Given that the machine learning algorithm has been trained with the dataset of Waring, that is likely underestimating the number of thermal springs in some regions (as suggested for New Zealand by the reviewer), it is hard to believe that the prediction will provide much more thermal springs than observed. Hence, we used the random-forest results to identify the regions where no thermal springs were reported at all. We added a caption to Figure S6.

But there is now so much more information: my googling “Tibet thermal springs chemistry” produced first a reference to a very recent catalog of over 200 springs, ~10x Waring's compilation; and as the second hit a catalog of 39 more springs in an area where Waring has none. [I had thought that perhaps the authors' reference to Keshi as a “large dataset on thermal springs” might fill the Chinese data gap, but this paper contains no listing of data, which leaves me wondering why was it cited?

We mentioned the incompleteness of Waring's dataset. Moreover, in the reviewed dataset we added an information on the number of thermal springs per site/area. In Tibet there are >40 thermal springs (in 3 areas in Tibet there are “several springs”). This information may reduce the incompleteness of the dataset and provide a better idea on the abundance of thermal springs in a region. Keshi shows a distribution of the geothermal springs in China that could be useful for geostatistical analysis.

Surely these statistics can be greatly improved by an additional literature search? Similarly the Minissale and the Ferguson-Grasby datasets (Figure S4) and the East African datasets (Figure S6) should be included in the main analysis. After all this, you can still make predictions of areas of missing data-sets, and now it will be more interesting and more useful!

In an initial stage of our work we replaced the North America and South Italy thermal springs with the dataset reported by Minissale and the Ferguson-Grasby. We then realized that we should have kept to substitute other thermal springs of Waring with other most recent dataset. This operation would represent another type of work, the onerous literature research that we hope our dataset would facilitate as guide line. So we decided to keep only Waring and, as suggested by the reviewer, we have now removed also the additional Cameroon-Svalbard-Deception.

The authors may argue that a complete literature search would be too onerous, and adding only the most easily found digital datasets would be capricious and statistically unwarranted: but why then do they add Cameroon-Svalbard-Deception to the already geographically biased compilation of Waring? Frankly, the real ‘danger’ is that Waring might prove to be essentially superseded, and all the excellent work the authors did to capture his data prove to be superfluous.

We perfectly understand the reviewer’s objection. The initial goal of this work was to make the work of Waring available as electronic format. We added the Cameroon dataset because we have studied this area in the past and we immediately noted the absence of geothermal springs. Since a complete literature is too onerous we decided to report only Waring so we removed from the dataset the Cameroon-Svalbard-Deception contribution. In supplementary material we mention the Cameroon dataset since the absence of thermal springs in this area in Waring is highlighted by the random forest algorithm.

First, local relief! Springs require a pressure differential to manifest at the surface. In principle this could be due to thermally created density differences, but water density changes by <5% between 0 and 100°C, and spring chemistry can increase density by ~4% (using sea-water end-member, Fig. 3). The driving pressure produced by a 1% density change over a 1km hydrostatic head is equivalently produced by <5m of topography! I predict that high local relief within a few km to a few tens of km will prove to be correlated with larger numbers of springs.

We thank the reviewer for this precious insight. In order to add the role of the local relief to the occurrence of thermal springs we calculated the topographic position index (TPI). TPI is a method of terrain classification where the altitude of each data point is evaluated against its neighborhood. If a

point is higher than its surroundings, the index will be positive, as for example on ridges and hilltops, while the figure will be negative for sunken features such as valleys. We obtained the calculated global TPI raster from Amatulli et al. (2018) with a resolution of 30-arc second (~1 km, <https://www.earthenv.org/topography>) and then calculated the TPI statistic for each hexagon and re-run the random forest algorithm. As the reviewer thought, the role of TPI/local relief is important in the thermal spring occurrence. For this purpose, we reviewed the R code for the random forest and noticed that we mistakenly took into account all the hexagons (water with no thermal springs + land with thermal springs). Re-running the code with only land provided a slightly different result with an important role of TPI and crustal thickness. Terrestrial heat flow is still the most important factor. We also added a processing that consider the number of thermal springs that occur in each geothermal site.

Second, incomplete re-equilibration of geothermal gradients in post-glacial time. Regions de-glaciated in the last 50ka will likely have temperatures decreasing with depth in the shallow sub-surface. I predict that regions still ice-covered at say 20 ka will prove to be correlated with lower numbers of thermal springs.

We thank the reviewer for this further useful suggestion. We searched a global dataset to add the de-glaciation effect among the factors influencing the thermal springs distribution. Richard Peltier is author and co-author of several works about modeling the ice thickness history of the last thousands of years. We selected his most recent model (ICE-7G, Roy & Peltier, 2018) that provide the ice thickness every 1 Ka year since 26 Ka. As suggested by the reviewer, we selected the ice-thickness at 20ka (not really different from 26 Ka) and added as parameter for our random forest analysis. Its variable importance is not among the top five (after Strike Slip regime there is a marked increase of the variable importance). We argue that the role of de-glaciation could be more important if analyzed by focusing on the northern hemisphere at a larger scale. Our analysis aims at identifying the geological factors influencing the thermal springs occurrence at a global and smaller scale.

Indeed, in Figure S6 (after western China discussed above), the next largest grey area of predicted under-reported springs is Scandinavia, and then areas of western Canada and Alaska's North Slope. Both are areas of late de-glaciation. And whereas googling New Zealand and Tibet brought up hundreds of springs missed by Waring, my equivalent search for Scandinavia produced ... nothing significant.

In Figure S6 we calculated the difference between predicted and observed number of thermal springs only where none were reported. Hence it is normal for New Zealand to have no grey areas. We understand that the prediction in the Scandinavian region is erroneous and we added this

consideration in the text. Unfortunately adding the de-glaciation dataset did not improve the prediction in these northern regions or only a little. We also understand that Waring does report for some regions only a fraction of the existing thermal springs

I urge you to re-run your random forest algorithm to test these two correlations, local relief and glacial history (of course, on the greatly expanded dataset I suggest you create).

We re-run the random forest algorithm with these two new datasets. We thank the reviewer for his precious insight. However we think that the purpose of this algorithm is to use a small dataset (we randomly selected about half of the available data for each of the 500 simulations) to understand the role each geological factor plays in the occurrence of the thermal springs calculating the variable importance.

I also suspect you are mis-interpreting the existence of “three main groups: warm, hot and boiling” (Figure 2a). I believe the peak above 90°C, and the sharp fall-off below 20° are biases in reporting. As I noted above, springs which are at local boiling point (or for which local boiling is reached in the very shallow sub-surface) will be visibly steaming and likely effusive or geysiring, so will be more widely noted particularly in early reports which are the basis of many of Waring’s data. Compilations by national surveys may prove to be more exhaustive and less biased. Springs with temperatures below 20°C are simply unlikely to be reported, except in the coldest climates. From the Pentecost reference the authors cite, “Waring (1965) noted that in Europe, thermal springs would normally be higher than about 20 °C and in the United States, above 15 °C”.

My expectation is that, above a lower temperature bound at which the catalog of springs is complete, the number of springs will decrease with increasing temperature, with the mathematical form of that decrease containing information about the relative difficulty of extracting water from different depths. If I am wrong, can you suggest any reason why there should be three separate modes of water temperature?

We thank the reviewer for initiating these stimulating discussions. We agree with the reviewer on the interpretation of the low-T and boiling springs populations. We suggest that the boiling springs are probably also affected by a clustering due to the buffer effect of the boiling process. We added a discussion on this possible distribution bias.

The .xls file has a lot of spurious false precision (temperatures to the nearest millionth of a degree; heat-flow accurate to 10 picowatts Watts/m2). Please display data appropriately! Lats

and longs from Waring to two decimal places, not to the nearest millimeter, with a footnote indicating that median accuracy is likely 0.1°

We corrected the number of decimals in table 1.

Table 1 – Heidback and Pasyanos references are reversed

Ok

Line 135: 14 km is ~0.125°, not 0.0002° of latitude!

Ok

Waring (1965) is alternately cited as 1976 (Table 1 and Figure 1); and 1995 (line 127 and 147)

Corrected

References to Tamburello, Le Marechal and Kusakabe are missing (and perhaps others, these were just the ones for which I looked).

We added Tamburello. Le Marechal and Kusakabe are no longer cited.

Line 67: what is the basis for stating that Waring’s maps are “very accurate”?

Correct objection. Removed

Georeferentiation should be georeferencing, throughout; neglectable becomes negligible.

Ok

Figures 6 and S7 seem to have at least one hexagon in an entirely water-covered area west of Alaska.

Corrected

Fig 1a – what are units of kernel density? I’m guessing #/km²? The color scales of Fig. 1 and Fig. 7 need to be made equivalent, not offset by orders of magnitude.

We have now calculated the density in Fig1a as number of springs per km². The logarithm base 10 of this density ranges from -5 to -2, the logarithm base 10 of the results of the model ranges -7 to -2. These 2 orders of magnitude of the lower scale represent the way of the model to fit the absence of thermal springs. We put both scale upper limits to -2.

Line 126 et seq. Results, and Line 177 et seq. Discussion are written implying the data plotted in Fig. 1 and Fig. 2 and Fig. 3 are from Waring alone – but in fact I am guessing they also include the Cameroon-Svalbard-Deception data. Please be explicit. Given how few of Waring’s data have chemistry, to what extent is Figure 3 dominated by the Cameroon-Svalbard-Deception data? How many data points are these?

We decided to remove the Cameroon-Svalbard-Deception data.

REVIEWER #2

It is stated in line 173 that gas discharge coordinates are shown in supplemental table data 1 but I did not see any information on gas discharge coordinates in this Table. Here it would be also useful to briefly mention if there are any data on the occurrence of gas discharges from any of these prongs. For example is there evidence for gas bubbles in the springs. If this is not available, in the current data set it could be pointed out that there is this gap in knowledge. Obviously it is beyond the scope of the paper to catalogue gas discharges from springs.

We checked the uploaded documents and it seems that supplementary table data 1 has been correctly uploaded as “358793_0_data_set_6365947_r8qjdk.xls”. Reviewer 1 successfully downloaded the data since he suggested to reduce the number of decimal places to 2-3. Remarks and comments on the thermal springs are provided by Waring only for some of them.

We carried out a text search looking for key words in Waring’s “Remarks and additional references” that can be linked to the occurrence of gas discharges (“fumarole/s” 137). In this way, we could add in the dataset a flag for the thermal springs that have fumaroles nearby. This comment helped us in identify other key words in Waring’s notes that have been counted for the reviewed dataset. In particular we refer to the status of “used for bathing” of a thermal springs and to the presence of a “resort” in the thermal springs. Those key words are important for understanding how much and where thermal springs are used for health and recreational purposes. We added a description of this result in the text.

We also extracted, when available, the number of thermal springs for each site. We noticed that we have missed to explain that the ~6,000 coordinates extracted from Waring actually represent geothermal sites and not just single thermal springs. For each site there might be up to 150 thermal springs. If we sum up the actual thermal springs for each site, Waring dataset counted ~12,500 thermal springs. We added this new parameter in the reviewed data set and a description in the revised manuscript.

Fig. 3d is very interesting. Authors state that peripheral waters are usually cool and dominated by deep sourced CO₂. However, in the diagram it appears that these waters have some of the highest temperatures. Likewise the volcanic waters for which we may expect high temperatures show some of the coolest temperatures. More explanation and clarification would be helpful when taking out this figure in the text.

Given the low number of water chemical compositions available, we think that further speculations on this dataset would be not appropriate. The purpose of this Figure is to show that the dataset includes different types of thermal waters

l. 29 'of' not 'about'

Ok

l. 51 Better "The aspect of electrical power production"

Ok

l. 53 clarify what risk is limited here.

Ok

l. 58 delete 'a'

Ok

l. 71 'map images'

Ok

l. 74 'allowed us to'

Ok

l. 81 'among which 750 are with..'

Ok

l. 96 better 'to be included in'

Ok

l. 131 negligible

Ok

l. 149 'values'

Ok

l. 149 'and lower values'

Ok

l. 227 'majority of '

Ok

REVIEWERS' COMMENTS

Reviewer #1 (Remarks to the Author):

I'm delighted to have provided my precious insights, and spotted one of the two most important controls on thermal spring distribution (TPI). My second idea (20ka ice thickness) was not useful (at least, not with the existing dataset that still woefully under-represents some key areas). Of course, this 50% success rate in my review only raises the question, what other possibilities have been missed?

Most important, is the authors' statement in their rebuttal, "But the difficulties we encountered have discouraged this work: this operation is really too onerous, it definitely needs much longer time".

I hate to be an old curmudgeon acting as a gatekeeper of science, but you have asked me to review this paper, and I will state my opinion:

If the authors are not willing to do the work, why should I be bothered to read the results? If Nature Communications wants to continue to develop a reputation for showcasing excellent science that couldn't quite make it into Nature (my impression when I read articles each month and think "wow, cool"), then this is not a paper for you. Maintain your standards, and encourage the authors to submit the revised version to a specialty journal like Geothermics - where I will still read the paper with interest: lower expectations for dramatic new insights, but still considerable interest.

Indeed, if Geothermics were to seek me out as a reviewer for this manuscript, I would undoubtedly encourage them to publish.

Reviewer #2 (Remarks to the Author):

In the revised version, the authors have adequately addressed the reviewers' comments and I suggest that the paper can now be published in its present form.

One of the main objections of the previous versions include comments regarding the completeness of the chosen spring data set to be evaluated in this paper. The authors explain why they have limited themselves to the Waring 1965 compilation. Their arguments make sense because, the data set provides consistency in information and if other data sets are included, these may be incomplete or contain incomplete information that may introduce additional biases and uncertainties. The authors now make clear in the manuscript which data sets are included in the compilation and give examples of the ones not included. As the authors state in l. 101, future work can build on this work and fill in the gaps with more recent data sets. However, this is beyond the scope of this work. What they have done (also suggested by a reviewer) was to further investigate the Waring data set for the number of springs per site, which further improves the revised version. The resulting compilation of springs and their relationships to geologic parameters is of significance, novel, useful to a broad audience in the geosciences and therefore warrants publication in Nat Com.

The value of this work is to analyze the Waring data set and examine correlations with other geologic parameters such as heat flow, topography, proximity to volcanic centers and tectonic faults. Even though the data set is limited, it still covers a vast area with about 12,000 springs. The findings are significant and clearly described in the revised version.

The authors also addressed the comment regarding the temperature distribution of the springs and provide additional insights and clarifications in the revised version. They have also adequately addressed the comments regarding topographic relief and recent glaciation, the latter of which has does not appear to significantly influence the spring locations.

In summary, this is important and novel work that is of interest to a wide community of geoscientists in academia and also in the applied field of geothermal exploration.

Point-by-point response to the reviewers' final comments

Global thermal spring distribution and relationship to endogenous and exogenous factors

REVIEWER #1

I'm delighted to have provided my precious insights, and spotted one of the two most important controls on thermal spring distribution (TPI). My second idea (20ka ice thickness) was not useful (at least, not with the existing dataset that still woefully under-represents some key areas). Of course, this 50% success rate in my review only raises the question, what other possibilities have been missed?

The topography has been a big oversight in our work, and we thank the reviewer again for this precious suggestion. We don't know what other possibilities we missed. In our work we propose a methodology and share a global dataset of thermal springs, the only one available so far, to find it out. The random forest application demonstrated how it could well assess the importance of geological factors in thermal waters occurrence. We are confident that this will be very useful for the scientific community.

Most important, is the authors' statement in their rebuttal, "But the difficulties we encountered have discouraged this work: this operation is really too onerous, it definitely needs much longer time". I hate to be an old curmudgeon acting as a gatekeeper of science, but you have asked me to review this paper, and I will state my opinion: If the authors are not willing to do the work, why should I be bothered to read the results? If Nature Communications wants to continue to develop a reputation for showcasing excellent science that couldn't quite make it into Nature (my impression when I read articles each month and think "wow, cool"), then this is not a paper for you. Maintain your standards, and encourage the authors to submit the revised version to a specialty journal like Geothermics - where I will still read the paper with interest: lower expectations for dramatic new insights, but still considerable interest. Indeed, if Geothermics were to seek me out as a reviewer for this manuscript, I would undoubtedly encourage them to publish.

We greatly appreciate that the reviewer states his opinion. It is the essence of the peer review process. We believe, especially after the first review process, that our paper offers new insights and important data for geothermal scientists and that Nat Comm provides the appropriate visibility for our work.

REVIEWER #2

In the revised version, the authors have adequately addressed the reviewers' comments and I suggest that the paper can now be published in its present form. One of the main objections of the previous versions include comments regarding the completeness of the chosen spring data set to be evaluated in this paper. The authors explain why they have limited themselves to the Waring 1965 compilation. Their arguments make sense because, the data set provides consistency in information and if other data sets are included, these may be incomplete or contain incomplete information that may introduce additional biases and uncertainties. The authors now make clear in the manuscript which data sets are included in the compilation and give examples of the ones not included. As the authors state in l. 101, future work can build on this work and fill in the gaps with more recent data sets. However, this is beyond the scope of this work. What they have done (also suggested by a reviewer) was to further investigate the Waring data set for the number of springs per site, which further improves the revised version. The resulting compilation of springs and their relationships to geologic parameters is of significance, novel, useful to a broad audience in the geosciences and therefore warrants publication in Nat Com.

The value of this work is to analyze the Waring data set and examine correlations with other geologic parameters such as heat flow, topography, proximity to volcanic centers and tectonic faults. Even though the data set is limited, it still covers a vast area with about 12,000 springs. The findings are significant and clearly described in the revised version.

The authors also addressed the comment regarding the temperature distribution of the springs and provide additional insights and clarifications in the revised version. They have also adequately addressed the comments regarding topographic relief and recent glaciation, the latter of which has does not appear to significantly influence the spring locations.

In summary, this is important and novel work that is of interest to a wide community of geoscientists in academia and also in the applied field or geothermal exploration.

We thank the second reviewer for his positive and flattering comments. We agree about the importance and novelty of our work and this is the reason of our submission to Nat Comm.